# ENTSUMV2: Data, Models and Evaluation for More Abstractive Entity-Centric Summarization

**Dhruv Mehra**[*1], **Lingjue Xie**[*1], **Ella Hofmann-Coyle**[*1]
**Mayank Kulkarni**[2†], **Daniel Preoţiuc-Pietro**[1]
[1]Bloomberg    [2]Amazon Alexa AI
dmehra19@bloomberg.net, lxie91@bloomberg.net, ehofmanncoyl@bloomberg.net

maykul@amazon.com, dpreotiucpie@bloomberg.net

## Abstract

Entity-centric summarization is a form of controllable summarization that aims to generate a summary for a specific entity given a document. Concise summaries are valuable in various real-life applications, as they enable users to quickly grasp the main points of the document focusing on an entity of interest. This paper presents ENTSUMV2, a more abstractive version of the original entity-centric ENTSUM summarization dataset. In ENTSUMV2 the annotated summaries are intentionally made shorter to benefit more specific and useful entity-centric summaries for downstream users. We conduct extensive experiments on this dataset using multiple abstractive summarization approaches that employ supervised fine-tuning or large-scale instruction tuning. Additionally, we perform comprehensive human evaluation that incorporates metrics for measuring crucial facets. These metrics provide a more fine-grained interpretation of the current state-of-the-art systems and highlight areas for future improvement.

## 1 Introduction

Controllable summarization is a rapidly expanding field of research that deals with creating summaries tailored to different elements (Fan et al., 2018; He et al., 2020; Hofmann-Coyle et al., 2022). The controllable elements include entities (Maddela et al., 2022), aspects (Amplayo et al., 2021; Ahuja et al., 2022), users' preferred style (Fan et al., 2018) and length (Kikuchi et al., 2016; Dou et al., 2021). Controllable summarization has the promise to increase the utility and usability of summarization systems by enabling users to obtain summaries that align with their specific needs and preferences (Maddela et al., 2022). Further, controllable summaries can be used in downstream applications like search (Varadarajan and Hristidis, 2006; Turpin et al., 2007), entity salience (Gamon et al., 2013; Dunietz

and Gillick, 2014), aspect-based sentiment classification (Pontiki et al., 2016) or question answering.

Abstractive summarization methods aim to produce new summaries (Nenkova et al., 2011), which can be obtained through selection, compression and reformulation of the given source document. Compared to extractive summarization, abstractive summarization can produce concise summaries that capture the essence of the source text using fewer words, making them more efficient for users to consume. However, abstractive summarization is prone to suffer from issues in consistency with the source document (or factual errors), coherence or fluency (Cao et al., 2018; Kryscinski et al., 2019; Lebanoff et al., 2019). To evaluate abstractive summaries, automatic metrics have been proposed, although their correlation with human evaluation on the desirable facts for a summary are not always high or consistent (Fabbri et al., 2021).

In this paper, we focus on the task of abstractive entity-centric summarization. Past research on this topic was limited by the ability to comprehensively evaluate models, relying either on single-faceted human quality judgments (Fan et al., 2018; He et al., 2020; Goyal et al., 2023) or reference entity-centric summaries which were very extractive (Maddela et al., 2022). To this end, we release an updated version of the ENTSUM dataset (Maddela et al., 2022), named ENTSUMV2, where summaries are deliberately made shorter and more abstractive. Moreover, we enhance the evaluation process of entity-centric summarization methods by incorporating a comprehensive multi-faceted human evaluation, specifically designed for this task. This human evaluation complements the standard automatic metrics, including ROUGE and BERTScore. By incorporating both automatic metrics and human evaluation, we aim to provide a thorough and robust evaluation of summarization model performance and show the path forward to improving models for this task.

---

[*]The authors contributed equally
[†]Work done while at Bloomberg

Separately, we explore training several model architectures on this task and propose several improvements to the training process, which substantially outperform the existing state-of-the-art (+2.5 BERTScore, +4.4 Rouge-L), instruction-tuned models and the strong entity-centric Lead3 heuristic.

## 2 Data

In this paper, we introduce the ENTSUMV2 dataset which contains more compressed abstractive summaries when compared to the original ENTSUM dataset. We build the ENTSUM dataset on top of The New York Times' summarization corpus (hereafter referred to as NYT) which is available to use via the LDC.[1] The dataset shares the same set of documents as ENTSUM, but with a stricter length constraint of up to 60 words, half of ENTSUM's. The annotations are performed by annotators trained over multiple rounds on a proprietary annotation platform. The annotators are presented with the original document, the target entity and the salient sentences for the target entity as annotated in the original ENTSUM dataset. A diagram with the annotation process is presented in Appendix A. For quality control, we additionally calculated the inter-annotator agreement for the EntSUMv2 dataset in the final training round using ROUGE-[1,2,L] and BERTScore between the abstractive summary and the proxy summary (entity salient sentences) provided to annotators at annotation time. The EntSUMv2 Krippendorff's alpha for ROUGE-[1,2,L] and BERTscore are 0.75, 0.84, 0.85 and 0.81 respectively, indicating a high overlap. We collect a single summary for each document and entity pair.

Table 1 displays summary statistics for the newly introduced ENTSUMV2 dataset in comparison to ENTSUM and other public datasets for summarization. There is a notable increase in the occurrence of novel n-grams compared to ENTSUM, albeit still less than other datasets like NYT or CNN/Daily Mail (Nallapati et al., 2016). Moreover, the average summary length in ENTSUMV2 is significantly shorter, with an average of 46 words compared to the 81 words in ENTSUM. This stricter length constraint presents a challenge for the model to effectively select the most essential information within the summarized output.

---

[1] https://catalog.ldc.upenn.edu/LDC2008T19

## 3 Methods

We experiment with several methods for abstractive summarization as follows:

### 3.1 Heuristics

**Lead3$_{ovr}$** is a generic summarization approach that disregards the target entity and simply selects the first three sentences from the document.

**Lead3$_{ent}$** selects the first three sentences in the document specifically mentioning the given entity following entity detection and coreference resolution, as described in ENTSUM (Maddela et al., 2022).

### 3.2 GSum

We start with GSum$_{ent-sent}$, an entity-centric summarization version of GSum (Dou et al., 2021) which obtained the best performance on abstractive summarization on ENTSUM (Maddela et al., 2022). GSum is a summarization framework that incorporates two encoders: one for the source document and another for the guidance signal. Our GSum setup closely follows the setup outlined in ENTSUM (Maddela et al., 2022), where the model weights are initialized with BART (Lewis et al., 2019) with a few modifications which we find lead to improved performance. First, we incorporate a dropout layer (p=0.5) into the guidance signal encoder stack of the model architecture. Second, we experiment with a two step training process (*two–step*) motivated by an analysis on the GSum results which showed the entity-centric model's ability to select key information from the source document could be improved. In Step 1, we train the GSum model to generate generic document summaries by providing only the source document and an empty guidance signal input. In Step 2, we load the best generic GSum summarization checkpoint and fine-tune with the entity guidance signal and proxy entity-centric summaries, as described in (Maddela et al., 2022), to produce entity-centric summaries.

### 3.3 T5

T5 (Raffel et al., 2020) is a transformer-based encoder-decoder model that is pretrained using text with dropped token sequences as input and the dropped out tokens delimited by their sentinel tokens as output. We fine-tune two base versions of the T5 model for entity-centric summarization. The first is T5-base, trained with a combination

| Dataset | Size | Avg. summary len. | | | Avg. article len. | | | % novel ngram | |
|---|---|---|---|---|---|---|---|---|---|
| | | sents. | words | char. | sents. | words | char. | unigram | bigram |
| NYT | 41265 | 4.9 | 117 | 677 | 36.9 | 1021 | 5471 | 11.5 | 39.5 |
| CNN/DAILY MAIL | 312085 | 3.7 | 56 | 297 | 33.1 | 782 | 3998 | 13.3 | 49.95 |
| ENTSUM | 2788 | 2.5 | 81 | 444 | 34.4 | 1002 | 5319 | 0.82 | 5.93 |
| ENTSUMV2 | 2788 | 1.8 | 46 | 251 | 34.4 | 1002 | 5319 | 1.69 | 10.72 |

Table 1: Comparison of the existing document summarization datasets with ENTSUM. We report the corpus size, average article and summary length (in terms of words, sentences, and characters), and percentage of novel n-grams in the summary when compared to the article.

of supervised tasks including summarization. The second is T5-v1.1-base, pretrained solely on the unsupervised objective, allowing us to assess its performance independently of mixed task fine-tuning or other summarization data. In addition, we investigate two training setups: we experiment with fine-tuning the model with proxy entity-centric summarization only (*proxy*), or train it in two steps (*two–step*), wherein we initially train the model to generate generic summaries and subsequently fine-tune it for proxy entity-centric summarization. The second approach aims to provide the models with additional contextual understanding through the first step of training.

### 3.4 Flan-T5

Large-scale instruction tuning using diverse NLP tasks has emerged as an alternative to single-task fine-tuning. We examine the efficacy of Flan-T5 (Chung et al., 2022), an enhanced version of the T5 model, which has undergone instruction-tuning using a wide range of tasks and instructions, including several summarization datasets such as CNN/Daily Mail (Nallapati et al., 2016), Gigaword (Rush et al., 2015), MultiNews (Fabbri et al., 2019), SamSum (Gliwa et al., 2019) and XSum (Narayan et al., 2018). To facilitate zero-shot entity-centric summarization, we employ prompt engineering techniques to guide the model in generating entity-centric summaries. We develop entity-centric summarization prompt templates, inspired by the Flan Collection templates[2] and explore two input strategies, as the model was not originally trained for the entity-centric summarization task. In the first, we provide the complete source document as input, and in the second only sentences containing the entity and its coreference are provided. In Appendix C, we present the performance evaluation of the Flan-T5 model across different prompts.

## 4 Experimental Setup

**Training Data** We employ proxy summaries from the NYT corpus during the fine-tuning process of GSum and T5 models for entity-centric summarization. The original corpus comprised 44,382 training and 5,523 validation pairs (document, summary) for generic summary. To generate entity-centric summaries, we select the first three sentences that mention the target entity. This selection is based on the entity recognition and coreference resolution methods as described in (Maddela et al., 2022). Given that each document in the corpus contained multiple entities, the training set expanded to 464,339 pairs, while the validation set grows to 58,991 pairs.

**Test Data** We use the ENTSUMV2 dataset for evaluation only, following (Maddela et al., 2022). We conduct experiments by splitting this dataset into training and test sets. However, training on this dataset, even when combined with the additional proxy summaries, does not result in any performance improvements. Therefore, to ensure more accurate and dependable evaluations of model performance, we utilize the entire ENTSUMV2 dataset exclusively for testing purposes.

**Implementation Details** The T5-base [3], T5-v1.1 [4], and Flan-T5-base [5] models are obtained from the HuggingFace model repository. We use the GSum implementation provided by the authors.[6] In our GSum experiments, we adhere to the hyperparameters and implementation details outlined in the GSum framework. We conduct fine-tuning of the T5 and T5 v1.1 models for 2 epochs with a learning rate of 2e-5. The batch size is set to 32, and the experiments are performed on Nvidia Tesla V100 GPUs. During inference, we impose a constraint

---

[2]https://github.com/google-research/FLAN

[3]https://huggingface.co/t5-base
[4]https://huggingface.co/docs/transformers/model_doc/t5v1.1
[5]https://huggingface.co/google/flan-t5-base
[6]https://github.com/neulab/guided_summarization

on the T5 models to limit the generated output to a maximum of 60 tokens.

# 5 Results

We evaluate all models using both automatic and human evaluation for a more comprehensive view on model performance.

## 5.1 Automatic Evaluation

The results of automatic evaluation are reported in Table 2. We employ the same set of automated metrics used in ENTSUM, namely ROUGE-1, ROUGE-2, ROUGE-L (Lin and Hovy, 2003) and BERTScore (Zhang et al., 2020). The results show:

- GSum and T5 based methods perform similarly in their best configurations, with GSum slightly better on BERTScore.
- The best performing summarization model outperforms the strong Lead3 entity-centric baseline on R-1 (+2.2), R-2 (+3), R-L (+2.2) and BERTScore (+0.9).
- Two step training on generic, then entity-centric summaries is beneficial, improving results on GSum. Since GSum takes in 2 inputs (source document and guidance signal) as opposed T5 which only takes a single input, we suspect that the two step training process acts like a curriculum based learning approach which helps the model learn more effectively. The GSum model first learns to summarize the overall key information from the provided source document. Then, it uses the additional provided signal to summarize the information relevant to the provided entity.
- Instruction-tuned models obtain decent results but only as part of a pipeline that selects the entity related sentences a priori. Otherwise, their performance is similar or lower to the Lead3 generic summary heuristic, showing they can not perform the entity control aspect.
- Both T5 and T5-v1.1 achieve comparable performance after being fine-tuned on proxy entity-centric summarization, despite T5-base being initially fine-tuned with multiple supervised tasks, including summarization. This shows that further training on out-of-domain summaries provides diminishing gains.
- We also compare the summarization models to oracle extractive summarization models that rely on identifying the Lead3 salient sentences (Lead3$_{ent}$ Salient) and Lead3 sentences used to write the summary (Lead3$_{ent}$ Sum-

mary) (Hofmann-Coyle et al., 2022). Despite evaluating on abstractive summaries, there remains a gap compared to these oracle extractive methods, highlighting that abstractive methods still need to be further enhanced to identify key entity information.

## 5.2 Faceted Human Evaluation

We conduct human evaluation of three top-performing methods of each type (GSum$_{two-step+dropout}$, T5-v1.1-base$_{two-step}$, Flan-T5-base$_{p2-entity}$) for a more comprehensive assessment. T5-v1.1-base is selected for a fair comparison with GSum as T5-base is trained with additional summarization datasets. The T5v1.1 output is restricted to the first 60 tokens for a fair evaluation, as it tends to produce longer summaries. Three independent raters evaluated all 480 summaries each. Summaries are ranked on a Likert scale of 1 to 5, with a focus on crucial aspects: entity-specificity (or relevance), factuality (or consistency), and fluency aligning with previous work on human evaluation for summarization (Kryscinski et al., 2019; Fabbri et al., 2021). We also include completeness, specifically for measuring the controllability aspect and an overall quality score. The evaluation guidelines are provided in Appendix E. The trained annotators achieve a Krippendorf Alpha (Krippendorff, 2011) of 0.48 with the authors on a random subset of 100 annotations. The inter-annotator agreement between annotators on the five aspects is 0.73. The agreement numbers are in line to past research (Fabbri et al., 2021). The facet based scores indicate that:

- GSum and T5 demonstrate divergent facet-level performance, notably on overall quality, despite similar overall ROUGE and BERTScore results. The different architectures of these models lead to distinct summary patterns, with GSum excelling in factuality and completeness.
- Despite the 10-point R-1 score difference between T5 and Flan-T5, the performance gap narrows in human evaluation. Flan-T5 is trained on a larger corpus and diverse tasks, which aid in sentence fluency but inhibit its performance in other areas due to the generic nature of its pretrained tasks. Additionally, both T5 and Flan-T5 struggle more with factuality, generating inaccurate or fictional information.
- All models are able to obtain controllability, al-

|  | ROUGE-1 | ROUGE-2 | ROUGE-L | BERTScore | Avg. Len Sent. / Word |
|---|---|---|---|---|---|
| **Heuristics** | | | | | |
| Lead3$_{ovr}$ | 28.8 | 15.1 | 25.6 | 55.3 | 3.0 / 99.38 |
| Lead3$_{ent}$ | 57.5 | 48.5 | 54.5 | 73.3 | 2.76 / 92.31 |
| **Abstractive Summarization Methods with Fine-tuning** | | | | | |
| GSum$_{ent-sent}$ | 55.1$_{0.17}$ | 47.0$_{0.19}$ | 52.3$_{0.19}$ | 71.7$_{0.11}$ | 3.05$_{0.21}$ / 99.66$_{0.21}$ |
| GSum$_{two-step+dropout}$ | 59.7$_{0.04}$ | **51.5$_{0.03}$** | **56.7$_{0.03}$** | **74.2$_{0.04}$** | 2.88$_{0.01}$ / 90.0$_{0.13}$ |
| T5-base$_{proxy}$ | 60.8$_{0.34}$ | 50.2$_{0.39}$ | 56.0$_{0.34}$ | 72.8$_{0.16}$ | 1.50$_{0.02}$ / 43.8$_{0.27}$ |
| T5-base$_{two-step}$ | **61.6$_{0.78}$** | 51.0$_{0.65}$ | **56.7$_{0.78}$** | 73.2$_{0.30}$ | 1.51$_{0.02}$ / 44.05$_{0.71}$ |
| T5-v1.1-base$_{proxy}$ | 61.5$_{0.24}$ | 51.1$_{0.26}$ | 56.7$_{0.28}$ | 73.1$_{0.11}$ | 1.55$_{0.01}$ / 43.70$_{0.16}$ |
| T5-v1.1-base$_{two-step}$ | 61.3$_{0.12}$ | 50.9$_{0.17}$ | 56.6$_{0.12}$ | 73.0$_{0.12}$ | 1.55$_{0.01}$ / 43.48$_{0.08}$ |
| **Abstractive Summarization Methods with Instruction-tuned Models and Zero-Shot Inference** | | | | | |
| Flan-T5-base$_{p2-ovr}$ | 25.3 | 11.9 | 21.6 | 54.6 | 1.24 / 35.0 |
| Flan-T5-base$_{p2-entity}$ | 52.1 | 40.9 | 47.8 | 69.8 | 1.09 / 32.2 |
| T5-base$_{entity}$ | 48.2 | 34.1 | 43.4 | 65.6 | 1.78 / 36.9 |
| **Methods using Oracle Entity Sentence Information** | | | | | |
| Lead3$_{ent}$ Salient | 62.8 | 55.4 | 59.6 | 76.3 | 2.73 / 91.31 |
| Lead3$_{ent}$ Summary | 69.6 | 63.5 | 66.6 | 80.4 | 2.53 / 86.0 |

Table 2: Automatic evaluation results of different summarization models on the ENTSUMV2 dataset. **Bold** typeface denotes the best performance overall and underlined numbers represent best performance within a class of methods. The fine-tuning results are averaged over 3 runs with different seeds and standard deviation is provided in the subscript.

| Model | Entity-Specificity | Factuality | Completeness | Fluency | Quality |
|---|---|---|---|---|---|
| GSum$_{two-step+dropout}$ | **4.85$_{0.5}$** | **4.69$_{0.62}$** | **3.71$_{0.82}$** | 4.17$_{0.48}$ | **3.17$_{0.67}$** |
| T5-v1.1-base$_{two-step}$ | 4.72$_{0.96}$ | 4.17$_{1.43}$ | 3.11$_{1.18}$ | 4.36$_{0.6}$ | 2.77$_{1.0}$ |
| Flan-T5-base$_{p2-entity}$ | 4.58$_{1.12}$ | 4.06$_{1.3}$ | 3.06$_{1.06}$ | **4.64$_{0.66}$** | 2.76$_{0.97}$ |

Table 3: Human evaluation results (average score$_{stdev}$) of three types of summarization models on a subset of the ENTSUMV2 dataset. **Bold** typeface denotes the best performance.

though Flan-T5 lags behind the other models, even if fed with sentences that contain the entity.

# 6 Conclusions

This paper presents a comprehensive analysis of abstractive entity-centric summarization. We introduce a new dataset - ENTSUMV2 with summaries that are more abstractive and almost half the length of the summaries in ENTSUM, posing additional challenges to summarization models. We explore different model types, improving upon previous top-performing models through data insights and training techniques, as well as surpassing the strong Lead3 entity-centric baseline. Finally, we conduct the first multi-faceted human evaluation on entity-centric summarization, revealing detailed insights into model behavior and trade-offs, suggesting potential avenues for further enhancement.

# Limitations

We only study the task of entity-centric summarization in English, as this is a relatively new task and there are no other datasets to build on with relevant and salient entity sentences selected, which we use as base for writing our summaries. Thus, the paper does not test the generalizability of our models and findings to other languages.

We train the model for a predetermined number of epochs without task specific validation as a validation dataset for entity-centric summarization is not available and we only use the entire ENTSUMV2 dataset for evaluation.

We limit our experimentation to the T5-base model due to its comparable number of parameters with the GSum model and due to limited compute resources. However, exploring the training of larger T5 models can provide valuable insights into the impact of model size on task performance.

We only use arguably the most popular metrics for automatic summarization (ROUGE, BERTScore). Using more metrics could provide a more complete picture of model performance.

# Acknowledgements

We would like to thank Rajarshi Bhowmik and the many members of Bloomberg AI group who provided invaluable feedback on this paper. We are grateful to our annotators for their diligence in per-

forming this annotation task and human evaluation of the summaries.

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

## A  Dataset Annotation Process

In Figure 1, we illustrate an example showcasing the multiple stages of annotation implemented in the ENTSUM dataset. The example encompasses four distinct stages of annotation. In this paper, our experiments focus on two particular stages: salient sentences and entity-centric summary. We use the entity-centric summary to evaluate the model performance. During the evaluation process, we compare the models' performance when provided with either the entire article or only the salient sentences as input.

## B  Qualitative Comparison of EntSUM and EntSUMv2

In Table 4, we illustrate the qualitative difference between the ENTSUM and ENTSUMV2 datasets. In ENTSUMV2 the abstractive entity-centric summaries are constrained to 60 tokens, resulting more abstractive and specific summaries. Entity-centric summaries in ENTSUMV2 are on average 33% shorter than the cooresponding summary in ENTSUM.

## C  Prompt comparison for Flan-T5

Table 5 compares the performance of the Flan-T5 model when selecting different prompts for entity-centric summarization. The evaluation of the prompts is conducted on the NYT validation dataset using proxy entity-centric summaries. Prompt 1 and Prompt 2 adopt the summarization prompts employed in Flan-T5 instruction-tuning,

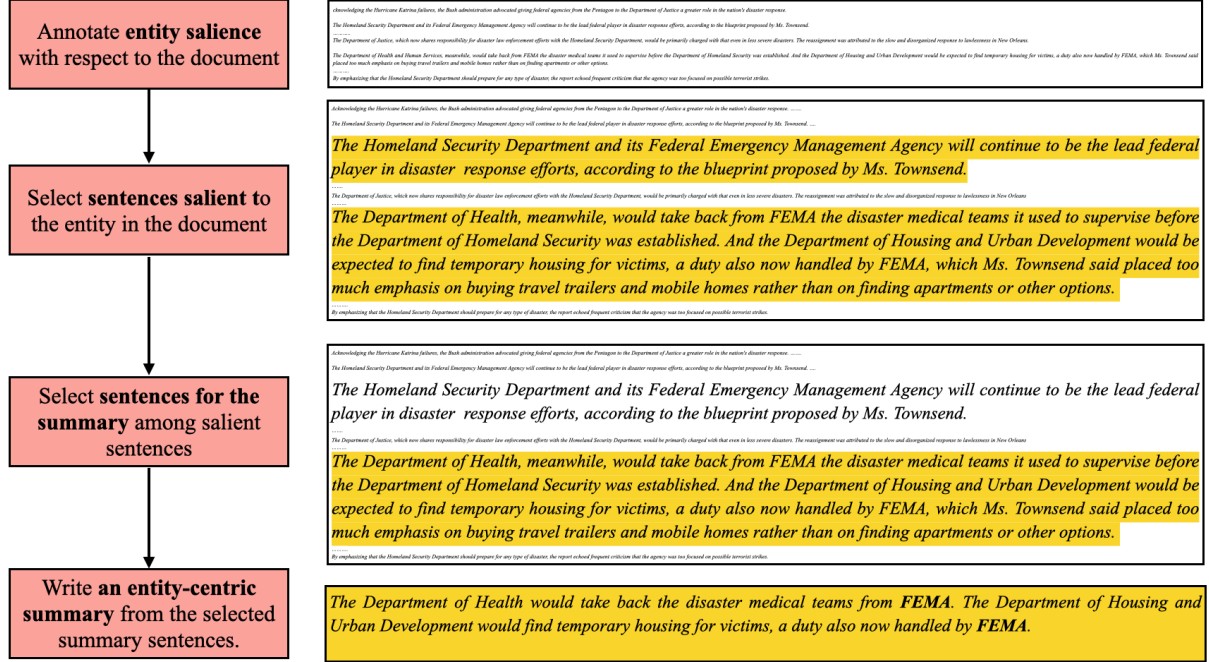

Figure 1: Annotation Pipeline as described in Maddela et al. (2022)

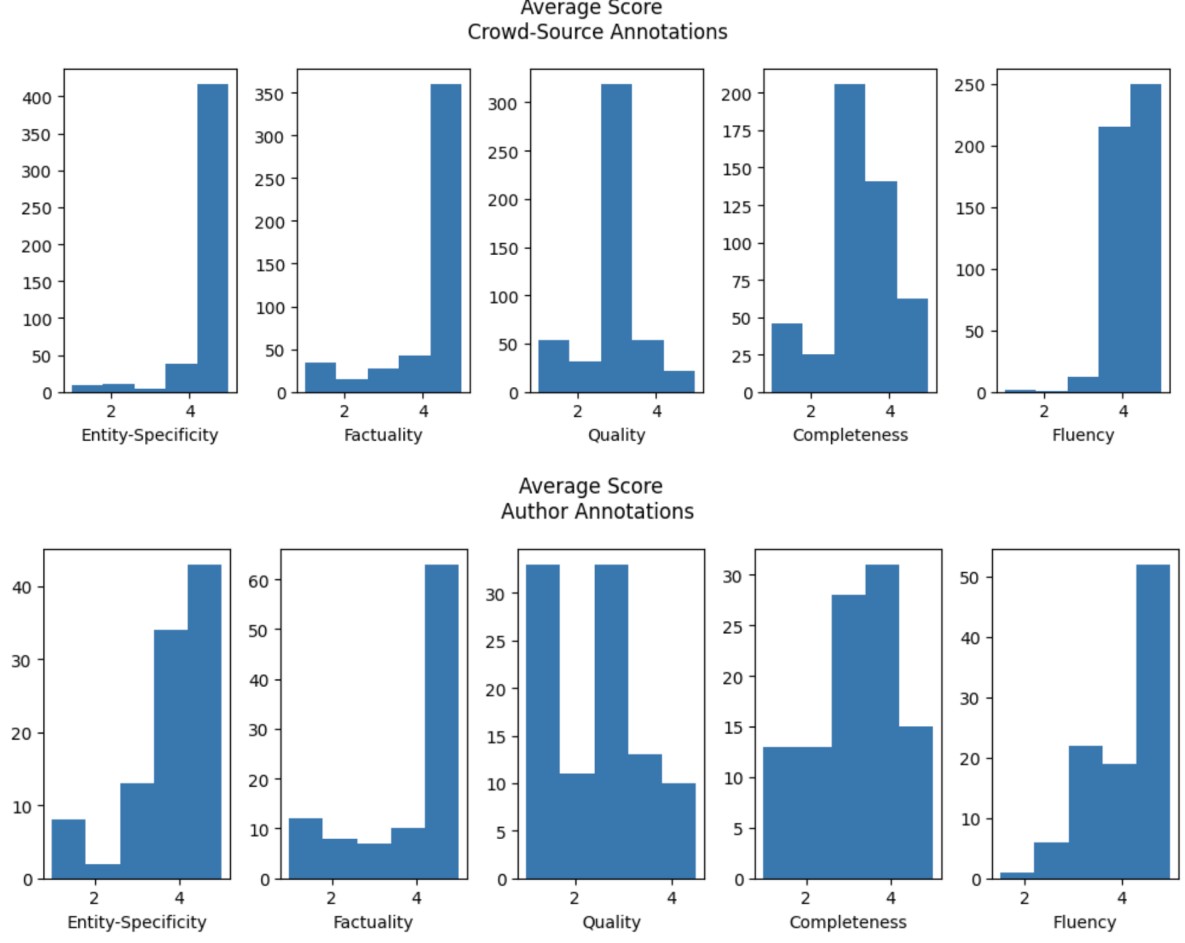

Figure 2: Histogram of average scores for trained annotator and author annotations, respectively

| Entity | EntSUM | EntSUMv2 |
|---|---|---|
| Bush | Focusing his priorities, President Bush invited ordinary people like a teacher, a physicist, an Afghan politician, the family of a fallen soldier to the State of the Union address. But a Democratic congresswoman turned the tables on Mr. Bush by inviting a guest of her own Cindy Sheehan, the antiwar protester who has determined Mr. Bush from his Texas ranch to the White House. When he entered the House chamber, his latest political trophy Samuel A. Alito Jr., newly confirmed and sworn in as a justice of the Supreme Court was on full display, a powerful reminder that Mr. Bush can still flex his muscles on Capitol Hill. | President Bush invited ordinary people to the State of the Union address. But a Democratic congresswoman turned the tables on Mr. Bush by inviting a guest of her own Cindy Sheehan, the anti-war protester who has determined Mr. Bush from his Texas ranch to the White House. |
| Jennifer Baker | The audience for the first day of the retrial, according to several people in the courtroom that day, included Mr. Giuca's mother, Doreen Giuliano, and Jennifer Baker, a young woman who had handed out pamphlets at Mr. Giuca's sentencing. She identified herself as a reporter for a weekly campus newspaper at Brooklyn College and said she was writing about the case. Annamaria Scaccia, editor-in-chief of The Kingsman, and Lauren Darson, managing editor of The Excelsior, said they did not employ anyone named Jennifer Baker and did not have reporters assigned to the courts or the district attorney's office. | Jennifer Baker, a young woman who had handed out pamphlets at Mr. Giuca's sentencing, said she was a student reporter from Brooklyn College. Annamaria Scaccia, editor-in-chief of The Kingsman, and Lauren Darson, managing editor of The Excelsior, said they did not employ anyone named Jennifer Baker. |
| Glen A. Rosenbaum | Glen A. Rosenbaum is a partner at the powerful law firm of Vincent & Elkins and spokesman for 18 top Texas law firms that have complained of inequities in the new taxing formula. While maintaining that they were willing to be taxed for the first time, Mr. Rosenbaum said, one way of making the plan fairer would be to raise the deduction per lawyer to at least $500,000 from the proposed $300,000. | Glen A. Rosenbaum is a partner at the powerful law firm of Vincent & Elkins and spokesman for 18 top Texas law firms that have complained of inequities in the new taxing formula. |
| Byun Ha Jung | Byun Ha Jung, a senior manager at the Hyundai Asan Corporation, the South Korean company and unit of the Hyundai Corporation that is developing the park, said that for South Korean companies, the reality is that one doesn't have to go to China. He asked reporters how much they have invested in China and whether it is one billion dollars. | Byun Ha Jung, a senior manager at the Hyundai Asan Corporation said that for South Korean companies, the reality is that one doesn't have to go to China. He asked reporters how much they have invested in China and whether it is one billion dollars. |

Table 4: Comparison of EntSUM and EntSUMv2

accompanied by additional entity-related information. Prompt 3 introduces an explicit word constraint. Prompt 4 adopts a question-answering style prompt, utilizing the 5W1H framework. The results show that the design of the prompts has a significant impact on the performance of the model. Prompts that closely resemble the task-specific prompts used during model training yield more accurate and relevant summaries. Prompt 2 is the selected prompt for the following evaluations.

## D Extended Human Evaluation Results

The results of the authors human evaluation results can be found in Table 6. The histograms of the trained annotators and author Likert scores for each facet are included in Figure 2.

## E Human Evaluation Guidelines

**Entity-Specificity**: for this metric we are determining to what extent the content pertains to the entity and is salient (relevant) in a summary about the entity. Please note the following:

- Please do not penalize the score for the entity name not being mentioned so long as the content still pertains to the entity.
- If all of the content pertains to the entity, but is not factually correct according to the source text, please score this metric 4 (All content is about the entity but the sentences may not be salient)

Scale anchors:

1. None of the content is about the entity
2. Most of the content is not about the entity
3. Some but not all of the content is about the entity
4. All content is about the entity but the sentences may not be salient
5. All content is about the entity and is salient

**Fluency**: this metric measures whether the summary is grammatically correct and easy to understand. Please do not penalize the score if the summary about the entity is incomplete (i.e., should include more details from the source text). The completeness metric measures this instead.

Scale anchors:

1. The summary is incomprehensible
2. Disfluent
3. Understandable
4. Good
5. Flawless

| Prompt | ROUGE-1 | ROUGE-2 | ROUGE-L | BERTScore |
|---|---|---|---|---|
| Summarize the following article focusing on {entity}: {text} | 34.2 | 23.3 | 30.6 | 61.7 |
| Write a short summary about {entity} based on the following article: {text} | 41.6 | 32.5 | 38.6 | 66.3 |
| Generate a summary under 60 words that describes {entity}, based on the following article: {text} | 38.3 | 27.7 | 34.6 | 63.8 |
| {text} \n Based on the article, answer the following question: {entity} did what to whom, when, where and how? | 13.7 | 10.2 | 13.3 | 47.7 |

Table 5: Comparison of Flan-T5 model performance on entity-centric summarization with different prompts.

| Model | Entity-Specificity | Factuality | Completeness | Fluency | Quality |
|---|---|---|---|---|---|
| $\text{GSum}_{two-step+dropout}$ | $\mathbf{4.75}_{0.62}$ | $\mathbf{4.66}_{0.66}$ | $\mathbf{3.71}_{0.83}$ | $4.13_{0.56}$ | $\mathbf{3.13}/0.73$ |
| $\text{T5-v1.1-base}_{two-step}$ | $4.59_{1.04}$ | $4.14_{1.48}$ | $3.07_{1.19}$ | $\mathbf{4.34}_{0.64}$ | $2.69_{1.05}$ |
| $\text{Flan-T5-base}_{p2-entity}$ | $4.48_{1.17}$ | $4.02_{1.33}$ | $3.04_{1.08}$ | $4.62_{0.7}$ | $2.7_{1.01}$ |

Table 6: Authors human evaluation results (average $\text{score}_{stdev}$) of three types of summarization models on a subset of the ENTSUMv2 dataset.**Bold** typeface denotes the best performance.

**Factuality**: this metric measures whether the summary is true to the source text. Please penalize the score if the summary introduces new facts that were not present in the source text.

Scale anchors:
1. Very untrue of the source text
2. Mostly untrue of the source text
3. Somewhat true of the source text
4. Mostly true of the source text
5. Very true of the source text

**Completeness**: this metric measures whether the summary includes a comprehensive overview of the source text that pertains to the entity.

Scale anchors:
1. Does not capture entity-specific or overall important information
2. Captures overall important information, but does not capture entity-specific information
3. Captures some entity-specific information
4. Mostly captures the entity-specific information
5. Completely captures the entity-specific information

**Overall Quality**: this measures, from a reader's point of view, whether a reader would be able to gain an overview of the essential information from the original source text that pertains to the entity if they did not have access to the original source and the entity name.

Scale anchors:
1. Poor
2. Fair
3. Good
4. Very good
5. Excellent