# OpenReview forum: "EntSUMv2: Dataset, Models and Evaluation for More Abstractive Entity-Centric Summarization"
_EMNLP/2023/Conference — EMNLP 2023 Main_

### Official Review · Reviewer_fMbs · 2023-08-03

**Soundness:** 4

**Excitement:**

4: Strong: This paper deepens the understanding of some phenomenon or lowers the barriers to an existing research direction.

**Paper Topic And Main Contributions:**

The authors contribute a new dataset for the task of entity-centric abstractive summarization. They evaluate the performance of different abstractive summarization models and perform detailed empirical evaluations and a crucial human evaluation study.

**Questions For The Authors:**

Please address Point 1 under "Reasons to reject".

**Reasons To Accept:**

1. The paper is well-written and easy-to-understand. The key details are well-explained.
2. The EntSumv2 dataset is an important contribution, and they evaluate a wide range of models using it.
3. The discussion of automatic evaluation results and faceted human evaluation is quite comprehensive and provides interesting insights.

**Reasons To Reject:**

1. The standard deviation values in Table 2 are not provided.

**Reproducibility:**

3: Could reproduce the results with some difficulty. The settings of parameters are underspecified or subjectively determined; the training/evaluation data are not widely available.

**Reviewer Confidence:**

4: Quite sure. I tried to check the important points carefully. It's unlikely, though conceivable, that I missed something that should affect my ratings.

---

> ### Author Rebuttal · Authors · 2023-08-28
>
> Thank you for your insightful review and feedback. We appreciate your suggestion and will incorporate your feedback to include standard deviation values to Table 2 in the final paper.

---

### Official Review · Reviewer_gpN3 · 2023-08-03

**Soundness:** 4

**Excitement:**

3: Ambivalent: It has merits (e.g., it reports state-of-the-art results, the idea is nice), but there are key weaknesses (e.g., it describes incremental work), and it can significantly benefit from another round of revision. However, I won't object to accepting it if my co-reviewers champion it.

**Paper Topic And Main Contributions:**

In this paper, authors focus on the task of abstract entity-centric summarization. They update a version of a well-known existing ENTSUM data set where summaries are deliberately made shorter and more abstractive.

**Questions For The Authors:**

Please answer the questions above.

**Reasons To Accept:**

The experiment setups looks fine. The idea of explicitly shortening the gold-standard summaries to make summaries more abstractive sounds good. Overall the performance improvement is also quite significant.

**Reasons To Reject:**

1. It is unclear how the annotators are trained and what was the overall agreement during the data creation stage (cf. line 98). Could you please elaborate on annotation guidelines and what was the overall

2. I am curious as to how much abstraction is still introduced. There is no analysis to show comparision on word-level or entity level b/w the summaries of two dataset. Please discuss more on this part as there needs to be some insight into how much the new data differs from old data.

3. The choice of models for faceted-human evaluation is not clear. Although it is discussed in text it is likely incorrect. As per the table T5-base should have been considere but it was not the case. Can you elaborate on this choice?

**Reproducibility:**

3: Could reproduce the results with some difficulty. The settings of parameters are underspecified or subjectively determined; the training/evaluation data are not widely available.

**Reviewer Confidence:**

2: Willing to defend my evaluation, but it is fairly likely that I missed some details, didn't understand some central points, or can't be sure about the novelty of the work.

---

> ### Author Rebuttal · Authors · 2023-08-28
>
> Thank you for your insightful review and feedback. Please see our responses to your questions below:
> 1.  Our annotation process followed the same steps and guidelines described in the original EntSUM paper and is additionally detailed in Appendix A of our paper. The summaries in the EntSUM v2 dataset were annotated by a single annotator on a proprietary platform (details on the platform would break anonymity but we will provide more information in the final version). To ensure quality of the annotations, we conducted three rounds of annotator training with data not included in the evaluation dataset and provided the most salient sentences about the entity along with the source document to the annotators.
>
> 2. The paper currently includes dataset-level statistics like average summary length between the two versions of the dataset and number of novel n-grams in the summary in Table 1, which provide insights as to how much EntSUMv2 differs from EntSUM. We can incorporate more comparison statistics across the two versions of the datasets, such as n-gram overlap, and also add qualitative analysis with examples in the final paper. We have additionally calculated the inter-annotator agreement for the EntSUMv2 dataset in the final training round using ROUGE-[1,2,L] and BERTScore between the abstractive summary and the proxy summary (entity salient sentences) provided to annotators at annotation time. The EntSUMv2 Krippendorff's alpha for ROUGE-[1,2,L] and BERTscore are 0.75, 0.84, 0.85 and 0.81 respectively, which we will include in the final paper.
>
> 3. Please find the explanation of our choice of models to evaluate in section 5.2. T5-v1.1-base was selected for a fair comparison with GSum, as T5-base is trained with additional summarization datasets whereas T5-v1.1-base was not.

---

### Official Review · Reviewer_d8Nz · 2023-08-04

**Soundness:** 4

**Excitement:**

4: Strong: This paper deepens the understanding of some phenomenon or lowers the barriers to an existing research direction.

**Paper Topic And Main Contributions:**

The paper proposes EntSUM-v2, a dataset for entity-centric text summarization. It revises the existing ENTSUM dataset (with extractive summaries) to include abstractive summaries. The paper finetunes different summarization systems for the task and benchmarks them on the dataset. Finally, the paper conducts the first multi-faceted human evaluation on entity-centric summarization, which reveals detailed insights into model behavior.

**Questions For The Authors:**

Are extractive summaries of the document provided to annotators while writing the abstractive summaries?

**Reasons To Accept:**

1) The paper creates a new dataset for entity-centric summarization which will be useful to the NLP community. Currently, there are no good abstractive entity-centric summarization datasets.

2) The paper proposed a new multi-faceted human evaluation setup for entity-centric summarization.

3) The paper is well-written and easy to understand. The experiment setup is sound and the experiments are described thoroughly.

4) The results provide good insights into the existing models.

**Reasons To Reject:**

1) Few annotations details are not clear. Is the dataset single-annotated or is more than one summary collected for each entity-document pair? Which platform was used to collect the data? Was any other quality control done other than training the annotators?

2) Will the authors make their code available? Otherwise, it will be difficult to replicate the results.

3) It is unclear from the novel ngram how much the abstractive summaries deviate from the extractive summaries. It would help to do more qualitative analysis and add examples from ENTSUM and ENTSUMv2 to understand the difference between the two datasets.

**Reproducibility:**

3: Could reproduce the results with some difficulty. The settings of parameters are underspecified or subjectively determined; the training/evaluation data are not widely available.

**Reviewer Confidence:**

4: Quite sure. I tried to check the important points carefully. It's unlikely, though conceivable, that I missed something that should affect my ratings.

---

> ### Author Rebuttal · Authors · 2023-08-28
>
> Thank you for your insightful review and feedback. Please see our responses to your questions below:
> 1. Our annotation process followed the same steps and guidelines described in the original EntSUM paper and is additionally detailed in Appendix A of our paper. The summaries in the EntSUM v2 dataset were annotated by a single annotator on a proprietary platform (details on the platform would break anonymity but we will provide more information in the final version). To ensure quality of the annotations, we conducted three rounds of annotator training with data not included in the evaluation dataset and provided the most salient sentences about the entity along with the source document to the annotators. We have additionally calculated the inter-annotator agreement for the EntSUMv2 dataset in the final training round using ROUGE-[1,2,L] and BERTScore between the abstractive summary and the proxy summary (entity salient sentences) provided to annotators at annotation time. The EntSUMv2 Krippendorff's alpha for ROUGE-[1,2,L] and BERTscore are 0.75, 0.84, 0.85 and 0.81 respectively, which we will include in the final paper.
>
> 2. We will attempt to release the code. Meanwhile we have provided specifics on the implementation details in Appendix E, and we will provide further model-specific details in addition to references to the open source libraries used in our experiments.
>
> 3. Thank you for the suggestion. The paper currently includes some dataset level statistics, such as average summary length and novel n-grams between for the two versions of the dataset in Table 1. We will do as suggested and augment the quantitative analysis to include additional comparison statistics and also add qualitative analysis with examples to the final paper.
>
> Regarding your question on whether extractive summaries are provided to annotators, we provided annotators with a reference of the sentences that were salient to the entity at the time of annotation, which serves as a proxy extractive summary. Please see Figure 1 in Appendix A for more details on the same. Thank you.

---

### Meta-Review · Area_Chair_3hFS · 2023-09-24

**Recommendation:** 4

**Metareview:**

The paper conducts the first multi-faceted human evaluation on entity-centric summarization, which reveals detailed insights into model behavior. And, all reviewers have expressed positive opinions about this paper. Therefore, I suggest this paper accepted as a short paper in the conference.

---

### Decision · Program_Chairs · 2023-10-07

**Decision:**

Accept-Main

**Comment:**

The paper conducts the first multi-faceted human evaluation on entity-centric summarization, which reveals detailed insights into model behavior. And, all reviewers have expressed positive opinions about this paper. Therefore, I suggest this paper accepted as a short paper in the conference.